

SciPost Phys. 1(1), 007 (2016)

# Role of fluctuations in the phase transitions of coupled plaquette spin models of glasses

**G. Biroli[1], C. Rulquin[2*], G. Tarjus[2] and M. Tarzia[2]**

**1** IPhT, CEA/DSM-CNRS/URA 2306, CEA Saclay, F-91191 Gif-sur-Yvette Cedex, France
LPS, Ecole Normale Supérieure, 24 rue Lhomond, 75231 Paris Cedex 05 - France
**2** LPTMC, CNRS-UMR 7600, Université Pierre et Marie Curie, boîte 121, 4 Pl. Jussieu, 75252
Paris Cedex 05, France

* rulquin@lptmc.jussieu.fr

## Abstract

We study the role of fluctuations on the thermodynamic glassy properties of plaquette spin models, more specifically on the transition involving an overlap order parameter in the presence of an attractive coupling between different replicas of the system. We consider both short-range fluctuations associated with the local environment on Bethe lattices and long-range fluctuations that distinguish Euclidean from Bethe lattices with the same local environment. We find that the phase diagram in the temperature-coupling plane is very sensitive to the former but, at least for the 3-dimensional (square pyramid) model, appears qualitatively or semi-quantitatively unchanged by the latter. This surprising result suggests that the mean-field theory of glasses provides a reasonable account of the glassy thermodynamics of models otherwise described in terms of the kinetically constrained motion of localized defects and taken as a paradigm for the theory of dynamic facilitation. We discuss the possible implications for the dynamical behavior.


# 1  Introduction

The plaquette spin models (PSM) [1–6] provide an interesting testing ground for theories of the glass transition. On the one hand, they are related to the $p$-spin interacting glassy systems that provide a basis for the Random First-Order Transition (RFOT) theory [7,8]. On the other hand, when the number of spins per plaquette, $p$, is equal to the number of plaquettes attached to a given spin, $c$, their dynamics on Euclidean lattices can be fully described by localized defects and the dynamic-facilitation theory [3,4,6,9]. In consequence, they provide a framework to understand the possible connection between these two theories.

In a recent series of papers [10–12], Garrahan, Jack and Turner studied such models with $p = c$ in dimensions $d = 2$ and $d = 3$, namely, the triangular plaquette model (TPM) with $p = c = 3$ and the square pyramid model (SPyM) with $p = c = 5$, respectively. In the phenomenology of glass-forming liquids these models represent "fragile" systems [13], for which the relaxation time $\tau$ grows with decreasing temperature $T$ in a super-Arrhenius manner [3,4,6,11], with $\log \tau \propto 1/T^2$. Garrahan and coworkers focused on the thermodynamic behavior found when coupling different copies of the system and when considering the similarity or overlap between configurations as an order parameter. They found strong numerical evidence for the existence of a transition line in the temperature $(T)$−coupling $(\epsilon)$ plane separating a low-overlap from a high-overlap phase and terminating in a critical point. This was obtained both in an "annealed" calculation, where two coupled replicas of the system evolve together, for the 2-d (TPM) and 3-d (SPyM) systems [10, 11] and in a "quenched" calculation, where the configurations of the system are biased to be similar to a fixed reference configuration, for the 3-d (SPyM) case [12].

The presence of such thermodynamic transitions between low- and high-overlap phases in the presence of some biasing field was first predicted on the basis of the mean-field models of glasses [14–17]. Signatures of the transitions were recently obtained in several computer simulations of 3-dimensional Lennard-Jones and hard-sphere glass-forming liquids [14, 18–26], albeit for rather small system sizes, and have then often been taken as indirect evidence

for the validity of the mean-field RFOT scenario. It is therefore puzzling to see a similar phenomenology in finite-dimensional plaquette spin models that have been considered as a paradigm for glass formation explained through purely dynamical arguments. What appears specific, though, about the thermodynamic transition line in these models is that it exactly goes to zero temperature at zero coupling $\epsilon$ (which then corresponds to the usual physical situation for glass formation) and does so in a singular manner [10–12]. This raises an interesting possibility, namely, that fluctuations present in finite-dimensional systems could generically depress the thermodynamic glass transition temperature $T_K$ predicted by the mean-field theory to zero temperature.

In this work we focus on the role of fluctuations in the thermodynamic behavior of plaquette spin models of glasses in the presence of a coupling between replicas of the system. To discuss the influence of the spatial fluctuations of the order-parameter field, here the overlap, we distinguish between long-range and short-range fluctuations. The distinction appears somehow arbitrary because fluctuations may of course appear on a continuum of scales. What we mean by "long-range fluctuations" are long wave-length fluctuations whose correlations in space may become scale-free, *e.g.*, near critical points. They are responsible for the difference between mean-field and finite-dimensional results at criticality or for the disappearance of metastability in finite dimensions; they can be present in (infinite) Euclidean lattices, but not in Bethe lattices and other tree-like or fully-connected structures in which the spatial correlations are intrinsically limited. "Short-range fluctuations" instead denote here fluctuations that are associated with the local environment, as, *e.g.*, the connectivity of the lattice, and that never become scale-free: such fluctuations are present in Bethe lattices (and Euclidean lattices as well of course) but are absent in the fully connected lattice which is typically a fluctuation-less system. Some of us have already stressed in a previous work [27] that the RFOT scenario is very fragile to the introduction of short-ranged fluctuations associated with a finite connectivity.

We show that the singular behavior of the transition line in the TPM and SPyM, with a transition temperature that goes to zero when the coupling goes to zero, is *not* the consequence of the long-range fluctuations, hence not an intrinsic property of finite dimensions. The very same behavior is found in the Bethe-lattice versions of the TPM and SPyM. In the SPyM case, the phase diagram predicted from the Bethe-lattice calculation, both in the annealed *and* in the quenched settings, is actually very similar to the 3-dimensional one obtained by Jack and Garrahan [12]: see Fig. 1. The location of the terminal critical point(s) is of course at higher temperature and higher coupling and the associated critical exponents are different, due to the absence of long-range fluctuations; similarly, as sketched in Fig. 1, one expects that these fluctuations will enforce convexity of the thermodynamic potential $V(q)$ (the free-energy cost for maintaining an overlap $q$ with a reference configuration [14, 28]), thus preventing true metastability. However, all this is akin to what is obtained in a conventional 3-dimensional ferromagnet at a first-order phase transition and does not call into question the qualitative or even semi-quantitative relevance of the mean-field description. (The influence of the long-range fluctuations is more severe for the TPM in the quenched calculation: the finite-temperature transition line which is found in the Bethe-lattice TPM does not exist in $d = 2$ as a consequence of the disorder-induced fluctuations.) We have also assessed the role of the short-range fluctuations by studying Bethe lattices with a connectivity $c \neq p$, which therefore do not exactly mimic the TPM or the SPyM at a local level. We find that the peculiar property of the phase diagram, with a transition temperature at $T_K = 0$ in zero coupling, is a specific feature of the case $c = p$ which is not valid otherwise: $T_K > 0$ for $c > p$ and $T_K$ is absent for $c < p$. A precise account of the local environment is therefore required to recover the main features of the SPyM phase diagram (and of the TPM one in the annealed description). The consequences of these findings for the understanding of the glassy dynamics will finally be discussed in the

conclusion.

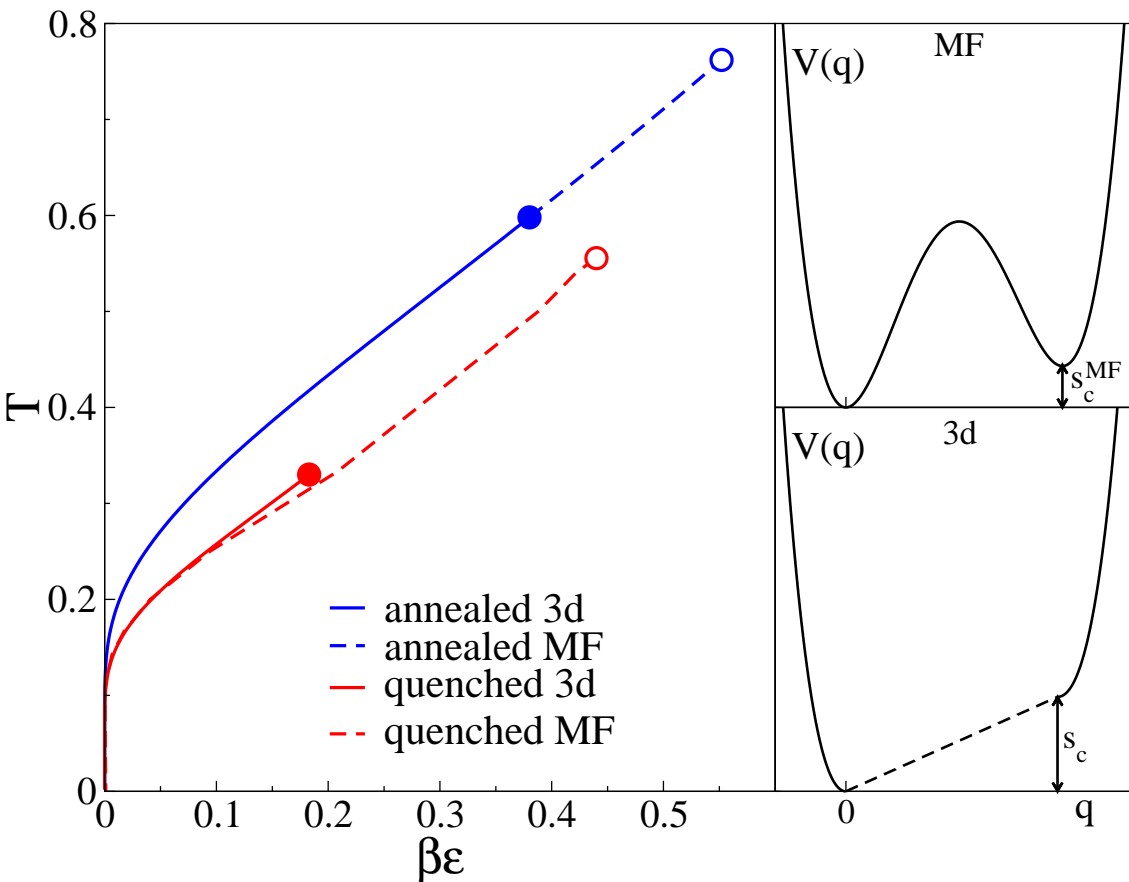

Figure 1: Temperature $T$ versus coupling $\beta\epsilon$ phase diagram of the square-pyramid model (SPyM) in both the annealed and the quenched descriptions for the Bethe hyper-lattice with $c = 5$ (dashed lines) and for $d = 3$ (full lines [11, 12]). The transition lines are between a high-$T$ phase with low overlap and a low-$T$ phase with high overlap. The interaction strength $J$ is set equal to 1. Inset: Sketch of the Franz-Parisi potential $V(q)$ as a function of the overlap in the mean-field description (left) and for $d = 3$ (right).

## 2  Plaquette spin models and overlap formalism

The Hamiltonian of the plaquette spins models reads

$$H[\mathscr{C}] = -\frac{J}{2}\sum_{\mu}\sigma_{\mu1}\cdots\sigma_{\mu p} \tag{1}$$

where $\mathscr{C} \equiv \{\sigma_i\}$ denotes the spins configuration on a lattice of $N$ sites, $J$ is a positive coupling, $\sigma_{\mu\alpha} = \pm1$, $\mu$ is the index characterizing the elementary plaquettes of the lattice, and $\alpha$ spans the $p$ sites around the plaquette.

We next introduce the so-called "overlap" between two configurations $\mathscr{C} \equiv \{\sigma_i\}$ and $\mathscr{C}' \equiv \{\sigma_i'\}$, which measures the similarity between them: the overlap at site $i$ is defined as $q_i = \sigma_i\sigma_i'$. As is clear from its definition, it is also an Ising variable, with $q_i = \pm1$.

As mentioned in the introduction, two settings, respectively called "quenched" and "annealed", have been introduced to characterize the glassiness of a given model. They both

involve studying the thermodynamics in the presence of an attractive coupling between configurations. We now present them in detail.

## 2.1 The "quenched" setting

In this case one is interested in the distribution of the overlap between the system's configurations and a fixed reference configuration $\mathscr{C}_0$ [14, 28]. The probability of a configuration $\mathscr{C}$ in the presence of an attractive coupling $\epsilon$ with $\mathscr{C}_0$ is then given by

$$p_\epsilon[\mathscr{C}|\mathscr{C}_0] = \frac{1}{Z_\epsilon[\mathscr{C}_0]} e^{\frac{\beta J}{2} \sum_\mu \sigma_{\mu 1} \cdots \sigma_{\mu p} + \beta \epsilon \sum_i \sigma_i \sigma_i^0}, \tag{2}$$

where $i$ and $\mu$ respectively denote the sites and the elementary plaquettes of the lattice and $\epsilon$ is the strength of the attractive coupling.

From the normalization factor $Z_\epsilon[\mathscr{C}_0]$ one defines the free energy of the system as a function of $\epsilon$: $W_\epsilon[\mathscr{C}_0] = \ln Z_\epsilon[\mathscr{C}_0]$ (it is rather $-\beta$ times the free energy). Due to the dependence on $\mathscr{C}_0$ this is a random function and it can be characterized by its cumulants, $W_1(\epsilon) = \overline{W_\epsilon[\mathscr{C}_0]}$, $W_2(\epsilon, \epsilon') = \overline{W_\epsilon[\mathscr{C}_0] W_{\epsilon'}[\mathscr{C}_0]} - \overline{W_\epsilon[\mathscr{C}_0]} \, \overline{W_{\epsilon'}[\mathscr{C}_0]}$, etc., where the overline denotes an average over the reference configuration. Although one could investigate the influence of a reference configuration equilibrated at a different temperature than the physical one $T = 1/\beta$, we focus here on the most relevant case where the reference configuration is drawn from the equilibrium Boltzmann distribution at temperature $T$:

$$p[\mathscr{C}_0] = \frac{1}{Z} e^{\frac{\beta J}{2} \sum_\mu \sigma_{\mu 1}^0 \cdots \sigma_{\mu p}^0}. \tag{3}$$

By a Legendre transform, one can define an effective potential and the average overlap with a reference configuration. The main quantity of interest is the Legendre transform of the average free energy $W_1(\epsilon)$,

$$\beta V(q) = -W_1(\epsilon) + \epsilon q, \text{ with } q = \partial W_1(\epsilon)/\partial \epsilon, \tag{4}$$

where $q$ now represents the average overlap; $V(q)$ corresponds to the mean thermodynamic cost to maintain a configuration at an overlap $q$ with a reference configuration and is usually called the Franz-Parisi potential [28]. In the case of mean-field systems, it encodes in a compact form interesting information, such as the configurational entropy, which is the difference in potential between the secondary and the main minimum, and the overlap of a typical metastable state sampled at equilibrium, which is also called Debye-Waller factor or non-ergodic parameter. It has been recently studied in several numerical works [14, 18–20, 22–24]. Whereas it stays nonconvex in finite-size systems in finite dimensions, it should have a convex shape as a function of $q$ in the thermodynamic limit (it is like a Helmoltz free energy) and it allows one to test to what extent the scenario obtained within mean-field models holds in finite-dimensional systems: see the sketch in Fig. 1 and the discussion in section 6 below.

## 2.2 The "annealed" setting

In this case one focuses on two coupled replicas $\mathscr{C} \equiv \{\sigma_i\}$ and $\mathscr{C}' \equiv \{\sigma_i'\}$, both equilibrated at the same temperature $T = 1/\beta$ with the Hamiltonian

$$H_\epsilon[\mathscr{C}, \mathscr{C}'] = H[\mathscr{C}] + H[\mathscr{C}'] - \beta \epsilon \sum_i \sigma_i \sigma_i', \tag{5}$$

where $H[\mathscr{C}]$ is given by Eq. (1). The free energy for the coupled replicas is defined as

$$W^{an}(\epsilon) = \ln \text{Tr} \exp(-\beta H_\epsilon[\mathscr{C}, \mathscr{C}']), \tag{6}$$

where the trace is over the spin variables $\sigma_i, \sigma_i' = \pm 1$.

## 2.3 Lattice models

Several different plaquette models have been studied in the literature, which are characterized by the number $p$ of spins around an elementary plaquette, the number $c$ of plaquettes attached to a given site, and more generally by the lattice type.

In the following we first consider cases with $c = p$. This includes in particular the models recently studied by Garrahan and coworkers [10–12] on 2-dimensional and 3-dimensional Euclidean lattices. In the TPM, the ferromagnetic interactions involve the 3 spins of each upward-pointing triangle in a triangular lattice and in the SPyM they involve the 5 spins of each upward-pointing square-based pyramid on a body-centered cubic lattice. In consequence, the TPM has $p = 3$ and each site of the lattice is connected to 3 triangles ($c = 3$) while the SPyM has $p = 5$ and $c = 5$. These models have been extensively investigated in relation to the theory of glass formation, more specifically for their connection to simple fragile glass-forming models with kinetic constraints [4, 6, 9–12, 29, 30]. Other models with $c = p$, but with an even number of spins per plaquette, such as the square-plaquette model on a square lattice ($c = p = 4$) or the cubic plaquette model on a cubic lattice ($c = p = 8$), have also been studied [1, 2, 6, 9, 29, 31]. Their dynamical behavior is related to that of kinetically constrained models of glasses, too, but, contrary to the TPM and the SPyM, they represent "strong" glass-formers [13] with an Arrhenius temperature dependence of the relaxation time, $\log \tau \propto 1/T$ at low $T$. Their behavior in the presence of a biasing field has not been investigated so far, and we will only briefly discuss them in the concluding remarks.

In order to discuss a mean-field version of the TPM and the SPyM we consider plaquette models on Bethe hyper-lattices (or Husimi trees, the analog of a Bethe lattice for systems with plaquette interactions [5]). These tree-like lattice structures have been widely used both in the context of the mean-field theory of structural and spin glasses and in computer science where they relate to the so-called XOR-SAT problem [5, 32]. In the present case, the lattice is formed of elementary plaquettes of $p$ sites that are connected through a tree-like structure with a fixed connectivity $c$. Each spin is involved in exactly $c$ plaquettes, hence in $c$ distinct $p$-spin interactions. Due to the tree structure, spatial fluctuations are restricted and the models have a mean-field character. In particular, they are known to be exactly described by the mean-field RFOT theory in the absence of coupling [5, 33–36].

# 3 Plaquette spin models with $c = p$: Generalities and existing results

## 3.1 From coupled replicas to plaquette models in a field

When $c = p$ the plaquette spin models have a dual representation in which one switches from the Ising spins, $\sigma_i$, defined on the sites of the original lattice with connectivity $c$ to the Ising plaquette variables, $S_\mu = \prod_\alpha \sigma_{\mu\alpha}$, placed on the dual lattice with the same connectivity $c$. As shown for instance in Refs. [3, 4, 6, 10], the mapping from one representation to the other is one-to-one with periodic boundary conditions (at least for the TPM and SPyM studied here). The correspondence is not exactly one-to-one for others boundary conditions, but is recovered in the thermodynamic limit.

In terms of the plaquette variables, one can reexpress the Hamiltonian in Eq. (1) as

$$H[\mathscr{C}] = -\frac{J}{2} \sum_\mu S_\mu, \tag{7}$$

which corresponds to a noninteracting Ising model in an external field $J/2$. As is well-known [3, 4, 6, 9], the dynamics is nonetheless glassy and the single-spin flip dynamics maps onto a

relaxation with kinetic constraints for the plaquette variables. The fact that $c$ plaquettes are connected to one and the same spin leads to this nontrivial dynamics. This representation in terms of plaquette variables is particularly useful to study the quenched and annealed Franz-Parisi potential.

We first rewrite the Hamiltonian of a coupled system for two configurations $\mathscr{C}$ and $\mathscr{C}_0$ in terms of the overlap variables $q_i = \sigma_i \sigma_i^0$:

$$e^{-\beta \mathscr{H}_\epsilon[\{q_i\}|\mathscr{C}_0]} = \sum_{\{\sigma_i = \pm 1\}} e^{\frac{\beta J}{2} \sum_\mu \sigma_{\mu 1} \cdots \sigma_{\mu p} + \beta \epsilon \sum_i \sigma_i \sigma_i^0} \prod_i \delta(q_i - \sigma_i \sigma_i^0). \tag{8}$$

Due to the properties of Ising variables, one has $\sigma_i = q_i \sigma_i^0$, and $\mathscr{H}_\epsilon[\{q_i\}|\mathscr{C}_0]$ can be expressed as

$$\mathscr{H}_\epsilon[\{q_i\}|\mathscr{C}_0] = -\frac{J}{2} \sum_\mu \sigma_{\mu 1}^0 \cdots \sigma_{\mu p}^0 q_{\mu 1} \cdots q_{\mu p} - \epsilon \sum_i q_i. \tag{9}$$

By using the dual representation for the configuration $\mathscr{C}_0$, we find

$$\mathscr{H}_\epsilon[\{q_i\}|\mathscr{C}_0] = -\frac{J}{2} \sum_\mu S_\mu^0 q_{\mu 1} \cdots q_{\mu p} - \epsilon \sum_i q_i. \tag{10}$$

We now consider separately the annealed and the quenched settings.

### 3.1.1 Mapping in the annealed case and self-dual line

We start with the simpler annealed case. The Franz-Parisi potential is obtained from the annealed free energy, which from Eqs. (5), (6), and (10) is given by

$$W^{an}(\epsilon) = \ln \sum_{\{q_i = \pm 1\}} \left( \sum_{\{S'_\mu = \pm 1\}} e^{\frac{\beta J}{2} \sum_\mu S'_\mu (1 + \prod_{\alpha=1}^p q_{\mu \alpha})} e^{\beta \epsilon \sum_i q_i} \right), \tag{11}$$

where the configuration $\mathscr{C}_0 \equiv \mathscr{C}'$ in Eq. (10) is now considered as annealed and we have therefore replaced the subscript 0 by a prime on $S_\mu$. By performing the sum over the plaquette variables explicitly, one ends up with

$$W^{an}(\epsilon) = \ln \sum_{\{q_i = \pm 1\}} e^{-\beta \mathscr{H}_\epsilon^{an}[\{q_i\}]}$$

where the effective Hamiltonian $\mathscr{H}_\epsilon^{an}[\{q_i\}]$ reads

$$\begin{aligned}
\mathscr{H}_\epsilon^{an}[\{q_i\}] &= -\frac{J}{2} \sum_\mu \ln[2 \cosh(1 + \prod_{\alpha=1}^p q_{\mu \alpha})] - \epsilon \sum_i q_i \\
&= -\frac{1}{2\beta} \ln[\cosh(\beta J)] \sum_\mu \prod_{\alpha=1}^p q_{\mu \alpha} - \epsilon \sum_i q_i + \mathrm{cst},
\end{aligned} \tag{12}$$

and cst denotes an irrelevant constant.

As first shown by Garrahan [10], the annealed computation therefore amounts to studying a plaquette spin model with a coupling $\tilde{J} = (1/\beta) \ln[\cosh(\beta J)]$ in a uniform external field $\tilde{H} = \epsilon$. This model is known to have an exact duality property [37–39], which implies that the partition function $Z(\tilde{J}, \tilde{H})$ associated with the Hamiltonian in Eq. (12) satisfies $Z(\tilde{J}, \tilde{H}) = [\sinh(\beta \tilde{J}) \sinh(2\beta \tilde{H})]^{N/2} Z(\tilde{J}', \tilde{H}')$ with $\tanh(\beta \tilde{J}'/2) = e^{-2\beta \tilde{H}}$ and $\tanh(\beta \tilde{H}') = e^{-\beta \tilde{J}}$.

As a result [10, 11], the annealed free energy $W^{an}(J, \epsilon)$, where we have made the dependence on the coupling $J$ explicit, satisfies

$$W^{an}(J, \epsilon) - \frac{N}{2} \ln[\sinh(2\beta\epsilon)] = W^{an}(J', \epsilon') - \frac{N}{2} \ln[\sinh(2\beta\epsilon')], \qquad (13)$$

where $\tanh(\beta J/2) = e^{-\beta\epsilon'}$ and $\tanh(\beta\epsilon/2) = e^{-\beta J'}$. There is therefore a self-dual line which is characterized by $\sinh(\beta J)\sinh(\beta\epsilon) = 1$. If the free energy has a singularity in a point $(\beta J, \beta\epsilon)$, by Eq. (13) it is also singular in the point obtained by the above transformation. As a result, if the model has a single phase transition, it must take place along the self-dual line which emanates from the point at zero temperature and zero coupling, $T = \epsilon = 0$. Note that the result is valid for the Euclidean lattices as well as for the Husimi trees, provided $c = p$.

### 3.1.2 Mapping in the quenched case

We now consider the quenched case. The reference configuration $\mathscr{C}_0$ in Eq. (9) represents some quenched disorder. More precisely, from the form of the Hamiltonian in Eq. (9) the disorder appears as random couplings, $JS_\mu^0$. Computing the quenched Franz-Parisi potential is then tantamount to obtaining the partition function of a model with random $p$-spin interactions in a uniform external field. The distribution of the random interactions are given by that of the variable $S_\mu^0$, which for an equilibrium distribution at temperature $T = 1/\beta$ is simply

$$p(\{S_\mu^0\}) \propto \prod_\mu \exp\left[\left(\frac{\beta J}{2}\right) S_\mu^0\right]. \qquad (14)$$

The average is given by $\overline{S_\mu^0} = \tanh(\beta J/2)$, whereas the variance is given by $\overline{S_\mu^0 S_\nu^0} - \left(\overline{S_\mu^0}\right)^2 = \delta_{\mu\nu}[1 - \tanh^2(\beta J/2)]$. Thus, the disorder is such that the average interaction is ferromagnetic and the fluctuations are smaller than the mean value, in particular at low temperature.

Another alternative formulation is also possible. One can switch from the overlap variables $q_i$ to the Ising overlap plaquette variables, $Q_\mu = \prod_{\alpha=1}^{p} q_{\mu\alpha}$, which leads to the expression $\mathscr{H}_\epsilon^{qu}[\{q_i\}|\mathscr{C}_0] = -\frac{J}{2}\sum_\mu S_\mu^0 Q_\mu - \epsilon\sum_i F(\{Q_{i\mu_i}\})$ where $F$ is a nonlinear function of the plaquette overlap [3]. This representation is not of practical use but it shows that the model is equivalent to a ferromagnet with a complicated many-body interaction in the presence of a random field $(J/2)S_\mu^0$ on the dual lattice. Finally, a simpler random-field model is found by going back to the site (Ising) variables with $\sigma_i = \sigma_i^0 q_i$: the model is a plaquette model with $p$-spin ferromagnetic interactions in the presence of a random field $\epsilon\sigma_i^0$. The free energy $W_\epsilon[\mathscr{C}_0]$ can be equivalently obtained within any of these representations. This however does not lead to further simplifications, such as the self-dual line found in the annealed case. To summarize, self-duality holds only in the annealed case for $c = p$.

## 3.2 Previous results: Phase transition in the $T - \epsilon$ plane for the TPM and SPyM in Euclidean space

In the annealed case, the two models, TPM and SPyM, both display a first-order transition line in the $T - \epsilon$ plane between a phase with a low overlap between the two replicas and one with a high overlap [10, 11]. This line terminates in a critical point that was found by Turner *et al.* in the universality class of the Ising model [11], as predicted by Franz and Parisi [40].

In the quenched case, for the 3-d SPyM, numerical simulations by Jack and Garrahan [12] gave evidence for the existence of a first-order critical line terminating in a critical point in the universality class of the random-field Ising model (RFIM), as theoretically predicted [40, 41]. On the other hand, there should be no finite-temperature phase transition for the TPM. Due

to the presence of quenched disorder, the existence of the first-order transition line and of a terminal critical point in the universality class of the Random Field Ising Model is excluded in two dimensions by the Imry-Ma argument [42, 43] and its rigorous formalization by Aizenman and Wehr [44].

In Appendix A we give heuristic arguments for why, as expected on general grounds [40, 41], the terminal critical point of the coupled plaquette spin models in the annealed setting is in the Ising universality class of the simple Ising model and that in the quenched setting in the universality class of the RFIM.

## 4 Bethe-lattice TPM and SPyM

We are primarily interested in the Bethe (hyper) lattice versions of the 2-dimensional TPM, which corresponds to $c = p = 3$, and of the 3-dimensional SPyM, which corresponds to $c = p = 5$. In these cases, the lattice is a Husimi tree in which each site is connected to exactly $c = p$ elementary plaquettes, themselves comprising exactly $p$ sites forming a triangle for the TPM and a pyramid for the SPyM: see Fig. 2 for the case $c = p = 3$.

As already mentioned, the mapping between site and plaquette variables, the mapping between coupled replicas and plaquette spin models in a field, and the duality relations hold also for Bethe hyper-lattices. They will be used in the following to simplify the analysis. Although we focus on models with $c = p$, it is also interesting to study cases in which $c$ is different from $p$ in order to discuss the results from a more general perspective. Along the way, we will therefore consider the treatment for generic values of $c$ and $p$.

### 4.1 Annealed case

When $c = p$, one expects that, if present, the phase transitions consist in a single first-order line terminating in a critical point. Thanks to the duality relation, one then knows that such a transition line must be along the *same* self-dual line as for the Euclidean case, which is defined by the relation $\sinh(\beta J) \sinh(\beta \epsilon) = 1$. Therefore, only the location of the terminal critical point is different. On general grounds one expects the mean-field, *i.e.*, Bethe-lattice, critical temperature to be higher than its Euclidean counterpart.

Furthermore, due to the tree structure of the Bethe hyper-lattice, one can derive self-consistent equations that allow one to obtain the phase diagram in the $T - \epsilon$ plane. As a result of the mapping to a plaquette model in a field (see above), the analysis is straightforward. By using the cavity method, a self-consistent recurrence equation on the cavity field $h$ can be found:

$$\tanh(\beta h) = \tanh\left(\frac{\beta \tilde{J}}{2}\right) \tanh\left[\beta(c-1)h + \beta\epsilon\right]^{p-1} . \tag{15}$$

After solving this equation and plugging the solution into the expression of the free energy of the plaquette model in a field one can reconstruct the phase diagram: see Appendix B for more details. Note that these cavity equations are valid for all values of $c$ and $p$.

### 4.2 Quenched case

In the quenched case, there is no self-dual line when $c = p$. Nevertheless, the cavity method also allows us to obtain the full phase diagram. This is again true for any values of $c$ and $p$. We have now to solve a plaquette spin model with random couplings in a field. The cavity method is more involved than in the annealed case, since we have to keep track of the whole cavity-field distribution. The cavity field $h_{\mu}^{(i)}$ represents the effect on site $i$ of plaquette $\mu$ of all

the spins except those involved in the plaquettes other than $\mu$ containing the site $i$ (see Fig. 2). The corresponding equations read

$$\tanh(\beta h_\mu^{(i)}) = \tanh\left(\frac{\beta J S_\mu}{2}\right) \prod_{j \in \mu \setminus i} \tanh\left[\beta \sum_{\nu \ni j \setminus \mu} h_\nu^{(j)} + \beta \epsilon\right] \tag{16}$$

where greek letters refer to plaquettes and latin letters to sites. The symbol $\setminus i$ means that site $i$ is excluded, and $S_\mu$ represents a quenched disorder (we have dropped the superscript 0 from the notations of the previous section) that can take the values $\pm 1$ with probability $e^{\pm \beta J/2}/[2\cosh(\beta J/2)]$. Fig. 2 provides a visual representation of the cavity fields. The above set of cavity equations leads to a self-consistent equation for the probability distribution $P(h)$. We then find the corresponding solution by using population dynamics [45–47], with a population of 10 millions fields. Finally, after plugging the solution in the expression for the free energy we obtain the phase diagram. More details are given in Appendix B.

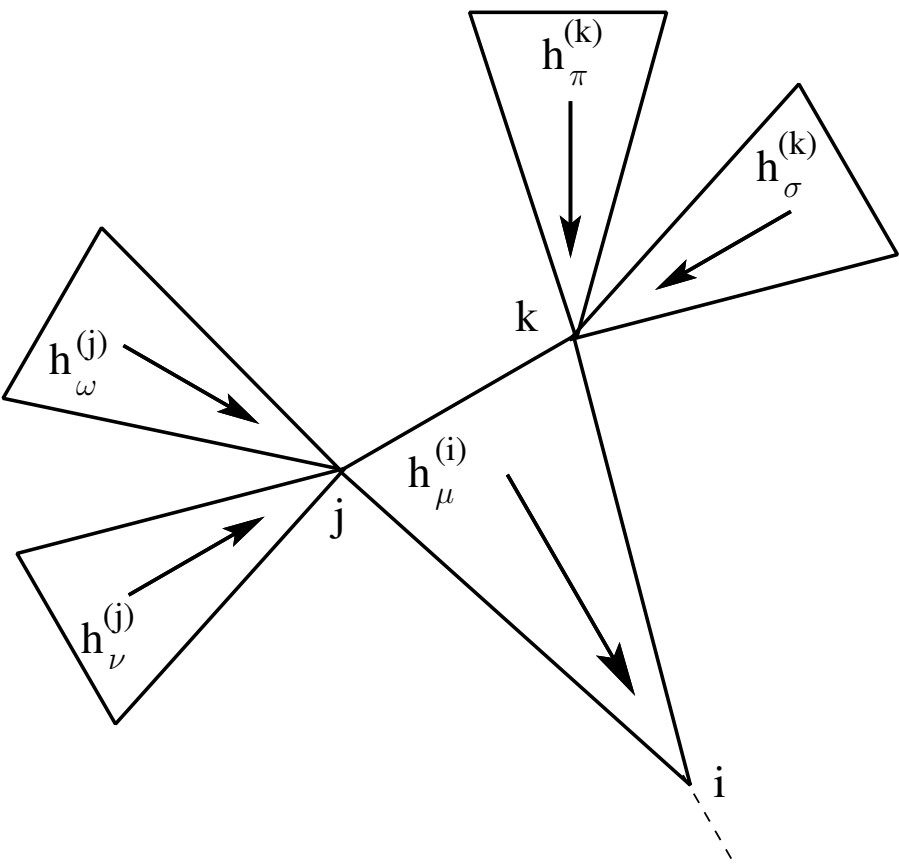

Figure 2: Local structure of the Bethe hyper-lattice for $p = c = 3$. The cavity fields that satisfy the recursive cavity equation in Eq. (16) are also shown.

## 4.3 Phase diagrams

The phase diagram that we obtain for the SPyM is shown in Fig. 1 and in the upper left panel of Fig. 3 while that for the TPM is shown in the left panel of Fig. 4. In both cases, the transition line (between a low- and a high-overlap phase) emerges from the singular point at $T = 0$ and $\epsilon = 0$. As anticipated, the transition line in the annealed case is on the self-dual line. It is

always above the quenched transition line, as could have also been expected since disorder suppresses the transition.

Both the annealed and the quenched transition lines $T_*(\epsilon)$ (or equivalently, $\epsilon_*(T)$, display a singular behavior, $\epsilon_* \sim T e^{-J/T}$, when $T \to 0$. This is due to the fact that the entropy and the configurational entropy vanish exponentially fast with the the temperature when $\epsilon = 0$.

Finally, the results obtained here for the SPyM and the TPM on Bethe hyper-lattices can be compared with those on Euclidean lattices. For the SPyM they are plotted together with the numerical results and estimates of Jack and Garrahan [12] in Fig. 1. The agreement is quite remarkable in this case. As expected for any mean-field treatment, the critical temperatures are overestimated compared to the $d = 3$ case, but the overall features of the phase diagram, including the singular behavior when $T$ and $\epsilon$ go to zero are similar in both descriptions. For the TPM, the annealed results are in good agreement between the Bethe hyper-lattice and the 2-dimensional (triangular) lattice. As already mentioned, this however can no longer be true for the quenched case: a transition is found on the Bethe lattice whereas it should be absent in $d = 2$. A more thorough discussion of mean-field versus finite-dimensional results will be given in the next section.

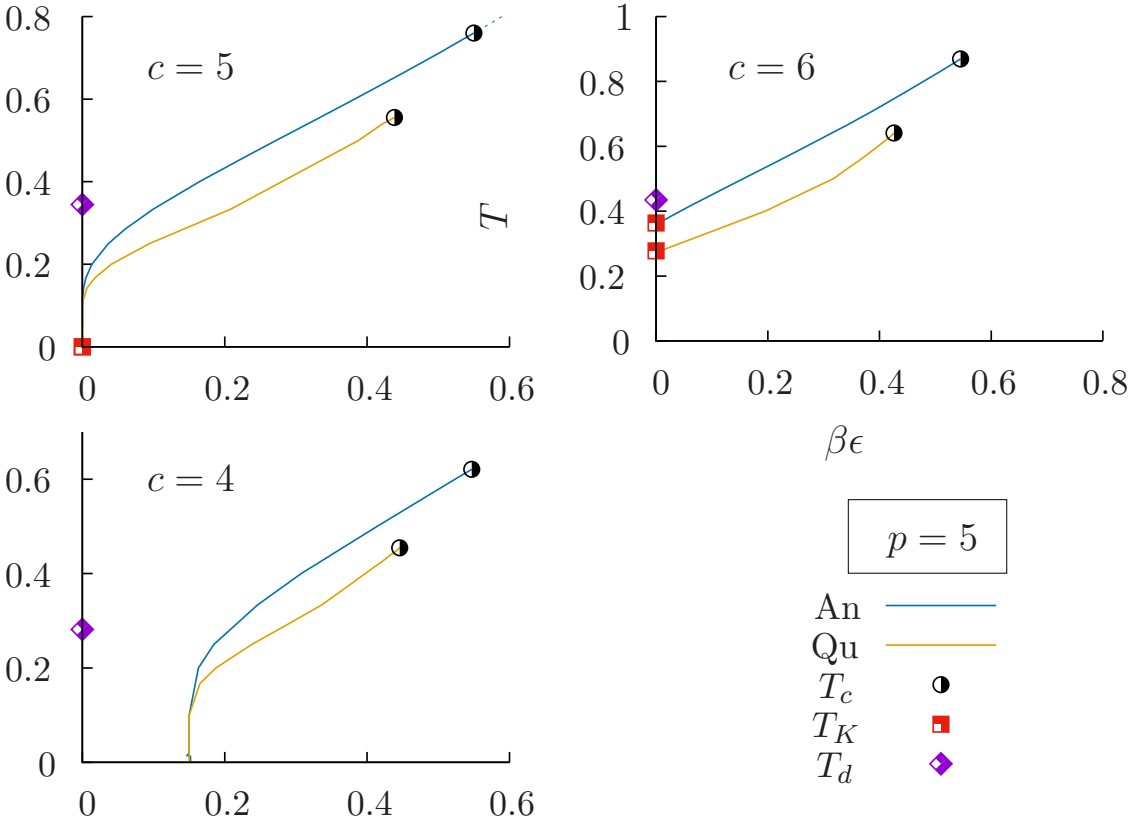

Figure 3: Temperature $T$ versus coupling $\beta\epsilon$ phase diagram for the plaquette spin models with $p = 5$ in both the annealed and the quenched descriptions on Bethe hyper-lattices with $c = 4, 5, 6$. The interaction strength $J$ is set equal to 1. The dashed line is the continuation of the self-dual line (see text). We consider $\beta\epsilon$ instead of $\epsilon$ for the horizontal axis to better compare all cases; otherwise the transition line for $c = 4$ terminates in $\epsilon_* = 0$ at $T = 0$, although the situation is quite different from the case $c = 5$ because there is a nonzero configurational entropy at $T = 0$ and $\epsilon_*$ actually scales as $T$ and not $T \exp(-J/T)$.

SciPost Phys. 1(1), 007 (2016)

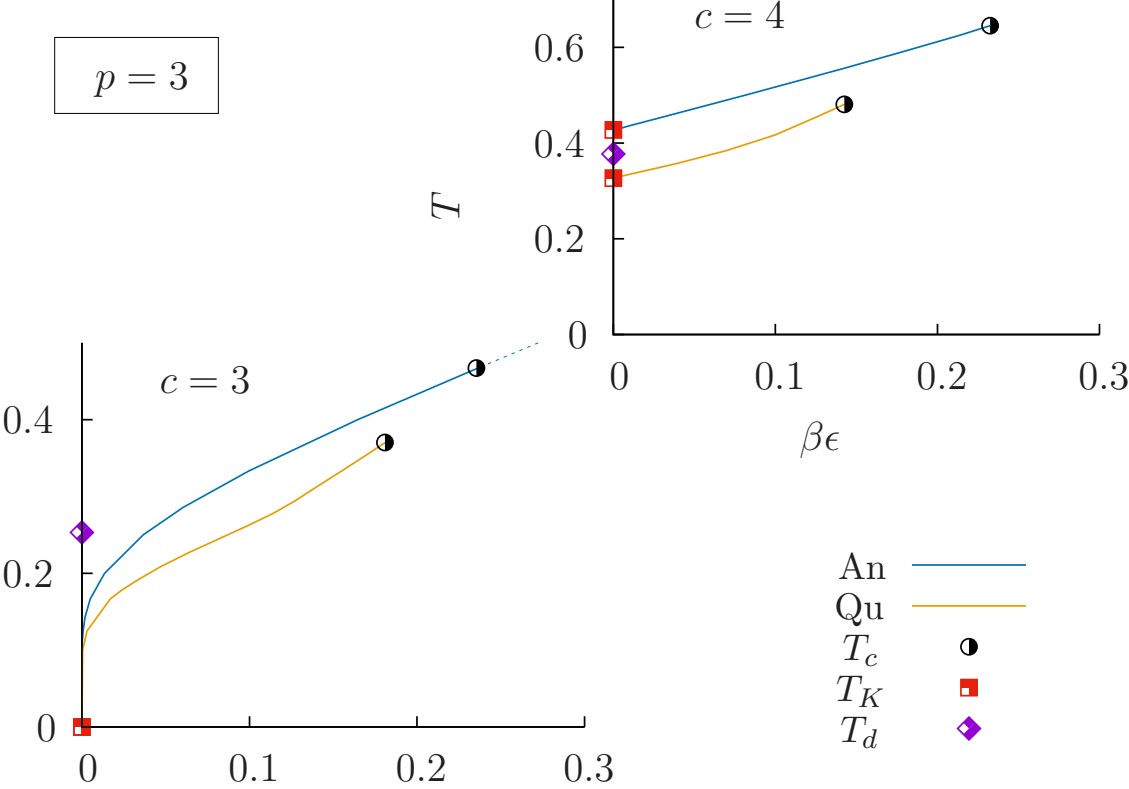

Figure 4: Temperature $T$ versus coupling $\beta\epsilon$ phase diagram for the plaquette spin models with $p = 3$ in both the annealed and the quenched descriptions on Bethe hyper-lattices with $c = 3$ and $c = 4$ ($c = 2$ corresponds to the 1-dimensional chain and is of no interest here). The interaction strength $J$ is set equal to 1. The dashed line is the continuation of the self-dual line (see text).

## 5 Role of fluctuations in glassy plaquette spin models

As discuss in the Introduction, our study puts us in a position to discuss separately the role of what we called "short-range fluctuations" and "long-range fluctuations" on the properties of glassy plaquette spin models.

### 5.1 Short-range fluctuations

One can probe the role of short-range fluctuations (which some of us previously argued to be important for the RFOT scenario [27]) by changing the connectivity $c$ at fixed $p$ for the Bethe hyper-lattice. In the absence of coupling $\epsilon$, it is known that the plaquette spin models on Bethe lattices can display, as temperature is lowered, a sequence of two transitions [5,33–36]: a dynamical one $T_d$ (akin to the transition predicted by the mode-coupling theory of glasses [48]), at which the system stays trapped in one of an exponentially large number of metastable states, and a static one $T_K < T_d$, at which the logarithm of the number of relevant metastable states (the complexity or configurational entropy) becomes sub-extensive. Below $T_K$, sometimes referred to as the "Kauzmann temperature", the system is in an ideal glass phase. This is a realization of the mean-field RFOT scenario. Adding a bias in the form of an attractive coupling $\epsilon$ with other configurations then produces a line of first-order transitions between a low- and a high-overlap phase that emanates from $T_K$ and terminates at higher $T$ and $\epsilon$ in a critical point.

For $c > p$ one finds that $T_K > 0$. The phase diagram in the $T$-$\beta\epsilon$ plane obtained from the cavity equations, is illustrated in the right panels of Figs. 3 and 4. (Note that the annealed transition line reaches the vertical axis at a temperature larger than $T_K$ but that may be either smaller or larger than $T_d$.) This is the conventional mean-field RFOT scenario, as it also appears in a fully connected lattice ($c \to \infty$) or in infinite dimensions. On the contrary, when $c < p$ the configurational entropy remains nonzero in $T = 0$ at $\epsilon = 0$ and there is therefore no Kauzmann transition in the absence of coupling. The transition line persists down to a threshold value $c_d$ of the connectivity but it reaches the zero-temperature line at a nonzero value of the scaled coupling $\beta\epsilon_*$ (see Fig. 3) [1].

The case $c = p$, which can also be realized in Euclidean space, appears somehow marginal: there is an entropy catastrophe but exactly at zero temperature, i.e., $T_K = 0$. The scaling behavior around $T_K$ is therefore expected to be significantly altered by this feature.

## 5.2 Long-range fluctuations

The role of long-range fluctuations can be assessed by comparing Bethe and Euclidean lattices with the same $c = p$. As seen from Fig. 1, long-range fluctuations do not change the topology of the phase diagram for the SPyM model, neither in the annealed setting nor, if one compares with the numerical results of Ref. [12], in the quenched one. They do of course modify the location of the transitions and change the behavior near the critical terminal points: the values of the critical exponent are the classical (mean-field) ones on the Bethe lattice but are those of the $d = 3$ pure or random-field Ising model in Euclidean space (see also Appendix A). Long-range fluctuations also enforce convexity of the potential $V(q)$ (see the sketch in Fig. 1) and prevent true metastability. As a result, the dynamical transition found at $T_d$ in $\epsilon = 0$, which can be associated to a spinodal point [where a secondary minimum first appears in the potential $V(q)$], is avoided and can at best remain in the form of a crossover.

For the 2-dimensional case in the quenched setting long-range fluctuations have a more severe influence. The TPM with $c = p = 3$ displays a transition line in the quenched calculation on the Bethe hyper-lattice but the transition should be absent on the triangular lattice. In Euclidean space, $d = 2$ is indeed the lower critical dimension of the RFIM [42–44], which means that the fluctuations depress the transition to zero temperature and zero random-field strength: no finite-temperature transition therefore exists in the $T - \epsilon$ plane.

# 6 Conclusion

This work is an assessment of the role of the fluctuations on overlap-based phase transitions in plaquette spin models of glasses in the presence of a biasing field. We have found that, at least for the 3-dimensional square pyramid model (SPyM) and when comparing with recent simulation results in Euclidean space [12], the mean-field description, provided it correctly encompasses the description of the local environment, appears surprisingly robust with respect to long-range fluctuations.

Needless to say, the $T - \epsilon$ phase diagram is informative but the most relevant aspect for glassy physics is the behavior for $\epsilon = 0$. The robust mean-field account of the transition line for the SPyM in $d = 3$ suggests that the system is well described as well in the absence of bias, as far as the thermodynamic aspects involving the overlap order parameter are concerned. Indeed, even though finite-dimensional fluctuations enforce convexity of the thermodynamic potential

---

[1]Both lines terminate at the same value of $\epsilon$. This is due to the fact that in the zero-temperature limit the couplings do not fluctuate any longer in the quenched case: they all become equal to $J$. In consequence, quenched and annealed settings coincide.

$V(q)$ describing the cost for maintaining an overlap $q$ with a reference configuration [28] and forbid true metastability, the glassy thermodynamics of the unbiased system in $\epsilon = 0$ should be correctly predicted by the mean-field theory up to a scale, the point-to-set length $\xi_{PTS}$, which diverges as $T \to 0$. [29] Moreover, as sketched in Fig. 1, if a transition is present in the presence of a biasing field $\epsilon$ in $d = 3$, $V(q)$ remains singular when $\epsilon = 0$ (with a linear segment) and a configurational entropy $s_c$ can then be univocally defined at low enough $T$, even in the thermodynamic limit and in finite $d$, in agreement with the assumptions underlying the RFOT theory [7,8]. It would be interesting to study how the point-to-set length diverges: although one expects $s_c \sim e^{-1/T}$ as $T \to T_K = 0$ in both mean-field and in $d = 3$, [12] the critical exponent characterizing the relation between $\xi_{PTS}$ and $s_c$ could be different.

Even more important, but more challenging, are the implications for the dynamics. On the one hand, it has been recently shown that the dynamics of a plaquette spin model on a random regular graph/Bethe lattice with $c = p$ (the authors focused on $c = p = 3$ but the conclusion is more general) [49] is equivalent to that of a kinetically constrained model of noninteracting spins. So, plaquette spin models with $c = p$ appear to share similar glassy features on Bethe and Euclidean lattices, even at the dynamical level. However, it is clear that the usual RFOT explanation of glassy dynamics in terms of mosaics and entropic droplets [7,8] does not apply for these models, for which the dynamics is instead ruled by the kinetically constrained motion of localized defects. There are two possible solutions to this apparent puzzle. One is that the RFOT theory gives a good description of the thermodynamics in three dimensions but that it fails for the dynamics. Another is that the RFOT arguments for the slow relaxation hold but that the proposed scalings [7] do not apply since the point-to-set length diverges at zero temperature only. The approach to a zero-temperature glass transition, which is not the usual situation envisaged by the RFOT theory, might lead to important changes (in fact metastable states seem to have zero surface tension in plaquette spin models [29]). Studying the plaquette spin model with $c = p = 8$ in $d = 3$, which behaves as strong glass-former, could provide some interesting insight on the connection between the thermodynamics associated with the overlap order parameter and the dynamics. In any case, elucidating this issue would lead to a substantial progress in the theoretical understanding of the glass transition problem.

## Acknowledgements

Support from the ERC grant NPRGGLASS and from the Simons Foundation (#454935, Giulio Biroli) is acknowledged.

## A  Universality class of the terminal critical points

If present the terminal critical points take place for nonzero values of the coupling $\epsilon$ and therefore nonzero values of the mean overlap $q$. As a result, one can expand the effective Hamiltonian for the overlap variables in the region around the critical point. It is convenient to move on to a soft-spin description by replacing the hard constraint $q_i = \pm 1$ by an additional term in the Hamiltonian $V(q_i) = (\lambda/8)(q_i^2 - 1)^2$ and let the $q_i$'s take any real value. The annealed and quenched Hamiltonians for the overlap variables become [see Eqs. (12) and (10) of the main text]

$$\mathcal{H}_\epsilon^{an}[\{q_i\}] = -\frac{\tilde{J}}{2} \sum_\mu \prod_{\alpha=1}^p q_{\mu\alpha} - \epsilon \sum_i q_i + \sum_i V(q_i), \tag{17}$$

$$\mathcal{H}_\epsilon^{qu}[\{q_i\}|\mathscr{C}_0] = -\frac{\tilde{J}}{2}\sum_\mu S_\mu^0 \prod_{\alpha=1}^p q_{\mu\alpha} - \epsilon\sum_i q_i + \sum_i V(q_i)\,, \tag{18}$$

with $\tilde{J} = (1/\beta)\ln[\cosh(\beta J)]$ and the $S_\mu^0$'s independently distributed variables with $\overline{S_\mu^0} = \tanh(\beta J/2)$, $\overline{S_\mu^0 S_\nu^0} = \tanh^2(\beta J/2) + \delta_{\mu\nu}[1-\tanh^2(\beta J/2)]$, etc. (see the main text).

For the annealed case, we simply expand around the saddle-point solution $q_*$: $q_i = q_* + \phi_i$, with $q_*$ solution of

$$-p\frac{\tilde{J}}{2}q_*^{p-1} - \epsilon + V'(q_*) = 0\,. \tag{19}$$

One then obtains

$$\mathcal{H}_\epsilon^{an}[\{q_i\}] - \mathcal{H}_\epsilon^{an}[q_*] = -\frac{\tilde{J}}{2}q_*^{p-2}\sum_{<ij>}\phi_i\phi_j + \frac{\lambda}{8}\sum_i[2(3q_*^2-2)\phi_i^2 + 4q_*\phi_i^3 + \phi_i^4] + \cdots, \tag{20}$$

where $< ij >$ is a sum over distinct nearest-neighbor pairs on the lattice and the ellipsis denote 3-body and higher-order ferromagnetic interactions. These interactions are known to be subdominant near the critical point if the pair interactions do not vanish, which is the case if $\epsilon_c > 0$, and consequently $q_* > 0$ (note that $q_*$ is the mean-field or saddle-point value and is different from the exact $q_c$, but this is irrelevant for the argument). The effective Hamiltonian in Eq. (20) has no $Z_2$ inversion symmetry, but as for the gas-liquid critical point of a fluid this is also known to be irrelevant at criticality (the $Z_2$ symmetry is asymptotically restored at the underlying renormalization-group fixed point). The critical point of the annealed model is therefore expected to be in the universality class of the Ising model.

For the quenched setting, the argument is slightly more involved. One expand the overlap variable as before, $q_i = q_* + \phi_i$; however, $q_*$ is the saddle-point solution not for the Hamiltonian in Eq. (18), which would be site-dependent due to the quenched disorder, but for the replicated theory,

$$\mathcal{H}_{\epsilon,\mathrm{rep}}^{qu}[\{q_i^a\}] = -\frac{1}{\beta}\sum_\mu \ln\left[\frac{\cosh[(\beta J/2)(1+\sum_a\prod_{\alpha=1}^p q_{\mu\alpha}^a)]}{\cosh(\beta J/2)}\right] \\ -\epsilon\sum_a\sum_i q_i^a + \sum_a\sum_i V(q_i^a)\,, \tag{21}$$

where $a = 1,\cdots,n$ is the replica index. Looking for a replica-symmetric and spatially uniform saddle-point solution leads to the following equation for $q_*$ when $n \to 0$:

$$-p\frac{J}{2}\tanh(\beta J/2)q_*^{p-1} - \epsilon + V'(q_*) = 0\,. \tag{22}$$

The effective hamiltonian can then be rewritten as

$$\mathcal{H}_\epsilon^{qu}[\{q_i\}|\mathscr{C}_0] - \mathcal{H}_\epsilon^{qu}[q_*|\mathscr{C}_0] = -\frac{J}{2}q_*^{p-2}\overline{S_\mu^0}\sum_{<ij>}\phi_i\phi_j - \frac{J}{2}q_*^{p-1}\sum_i\sum_{\mu/i}(S_\mu^0 - \overline{S_\mu^0})\phi_i \\ -\frac{J}{2}q_*^{p-2}\sum_{<ij>}\sum_{\mu/<ij>}(S_\mu^0-\overline{S_\mu^0})\phi_i\phi_j + \frac{\lambda}{8}\sum_i[2(3q_*^2-2)\phi_i^2 + 4q_*\phi_i^3 + \phi_i^4] + \cdots, \tag{23}$$

where $\overline{S_\mu^0} = \tanh(\beta J/2)$, the sum on $\mu/i$ is over all plaquettes attached to site $i$ and that on $\mu/ < ij >$ is over all plaquettes sharing the edge $(ij)$; the ellipsis denotes 3-body and higher-order interactions. The Hamiltonian is therefore that of a lattice scalar-field theory with random fields, $(J/2)q_*^{p-1}\sum_{\mu/i}(S_\mu^0-\overline{S_\mu^0})$, and random bonds, $(J/2)q_*^{p-2}\sum_{\mu/<ij>}(S_\mu^0-\overline{S_\mu^0})$. Provided $q_* > 0$, the dominant features at long distance are thus the ferromagnetic pair interactions and the random field, and the associated critical point is then expected to be in the universality class of the RFIM.

# B Cavity equations for the Bethe-lattice coupled plaquette spin models

## B.1 Cavity equations

As represented in Fig. 2, one can define an effective field $h_\mu^{(i)}$ representing the effect on site $i$ of plaquette $\mu$ of all the spins, when all other plaquettes containing site $i$ have been removed. We recall that greek letters refer to plaquettes and latin letters to sites.

The basic assumption is that the different cavity fields of a Husimi tree are uncorrelated in the large-size limit. Typical loops are indeed of order log(size) and have a vanishing contribution when the system size goes to infinity. The recursive structure of the lattice allows one to write iterative equations for cavity quantities. One sub-tree of connectivity $c$ can be constructed from a plaquette of $p$ sites by adding $(p-1)(c-1)$ branches emanating from $p-1$ sites and by leaving unconnected one "cavity" site. In turn, $(p-1)(c-1)$ such sub-trees, whose cavity fields are known, can be attached to a new plaquette $\mu$ to form a new, larger, sub-tree (Fig. 2).

For one configuration $\{q_\mu^{(1)}, ..., q_\mu^{(p)}\}$ of the plaquette $\mu$, the energy $E_\mu(\{q_\mu^{(1)}, ..., q_\mu^{(p)}\})$ of the new sub-tree can be written as the sum of the contributions due to the individual effective cavity fields acting on sites $\{1, ..., i-1, i+1, ..., p\}$ of plaquette $\mu$ plus the energy of the plaquette in the presence of a constant external field $\epsilon$ (not counted on site $i$):

$$E_\mu(\{q_\mu^{(1)}, ..., q_\mu^{(p)}\}) = -\frac{J}{2} S_\mu \prod_{j\in\mu} q_\mu^{(j)} - \sum_{j\in\mu\setminus i} (\epsilon + H_\mu^{(j)}) q_\mu^{(j)} \tag{24}$$

where $H_\mu^{(j)} = \sum_{\nu\ni j\setminus\mu} h_\nu^{(j)}$ is the total cavity field acting on site $j$. The notations $\setminus i$ and $\setminus\mu$ mean that we exclude the spin $i$ or the plaquette $\mu$ from the sum.

We consider first the quenched case. Then, $S_\mu = \pm 1$ is a binary random variable taken from the distribution $p[S_\mu = \pm 1] = e^{\pm\beta J/2}/[2\cosh(\beta J/2)]$. Tracing out over the configurations $\{q_\mu^{(1)}, ..., q_\mu^{(i-1)}, q_\mu^{(i+1)}, ..., q_\mu^{(p)}\}$ gives the effective cavity field $h_\mu^{(i)}$ acting on site $i$ of the plaquette $\mu$ via

$$C e^{\beta h_\mu^{(i)} q_\mu^{(i)}} = \sum_{\{q_\mu^{(1)}, ... q_\mu^{(p)} \setminus q_\mu^{(i)}\} = \{\pm 1\}} e^{-\beta E_\mu(\{q_\mu^{(1)}, ..., q_\mu^{(p)}\})} \tag{25}$$

with $C$ a normalization constant. Eq. (25) can be further rewritten as [5]

$$\tanh(\beta h_\mu^{(i)}) = \tanh(\beta J S_\mu/2) \prod_{j\in\mu\setminus i} \tanh\Big[\beta \sum_{\nu\ni j\setminus\mu} h_\nu^{(j)} + \beta\epsilon\Big]. \tag{26}$$

The $h_\mu^{(j)}$'s are random variables that depend on the disorder realization. Therefore, we have to follow their whole probability distribution $P(h)$. In the thermodynamic limit, Eq.( 26) becomes a self-consistent integral equation for $P(h)$:

$$P(h) = \sum_{S_\mu = \pm 1} p[S_\mu] \int \prod_{j=1}^{c-1} \prod_{\nu=1}^{p-1} \Big[dh_\nu^{(j)} P(h_\nu^{(j)})\Big] \delta\Big(h - h_\mu^{(i)}\Big) \tag{27}$$

with $h_\mu^{(i)}$ given by Eq.(26). This equation can be solved numerically by using population dynamics [45–47]. Our population has a size of 10 millions fields.

In the case of annealed setting, the quenched disorder is absent and the plaquette interaction coupling is fixed to $\tilde{J} = (1/\beta)\ln[\cosh(\beta J)]$, as detailed in the main text. Therefore $P(h)$

converges to a Dirac delta function and we have to solve a simple algebraic iteration equation for the cavity field,

$$\tanh(\beta h) = \tanh(\beta \tilde{J}/2) \tanh\left[\beta(c-1)h + \beta \epsilon\right]^{p-1} \tag{28}$$

## B.2 Free energy

In order to compute the free energy per site one starts with $p(c-1)$ sub-trees. As illustrated in Fig. 5, one can either add a new plaquette $\mu$ and connect $(c-1)$ sub-trees to each site of the plaquette or add $(p-p/c)$ sites linked to $c$ sub-trees.

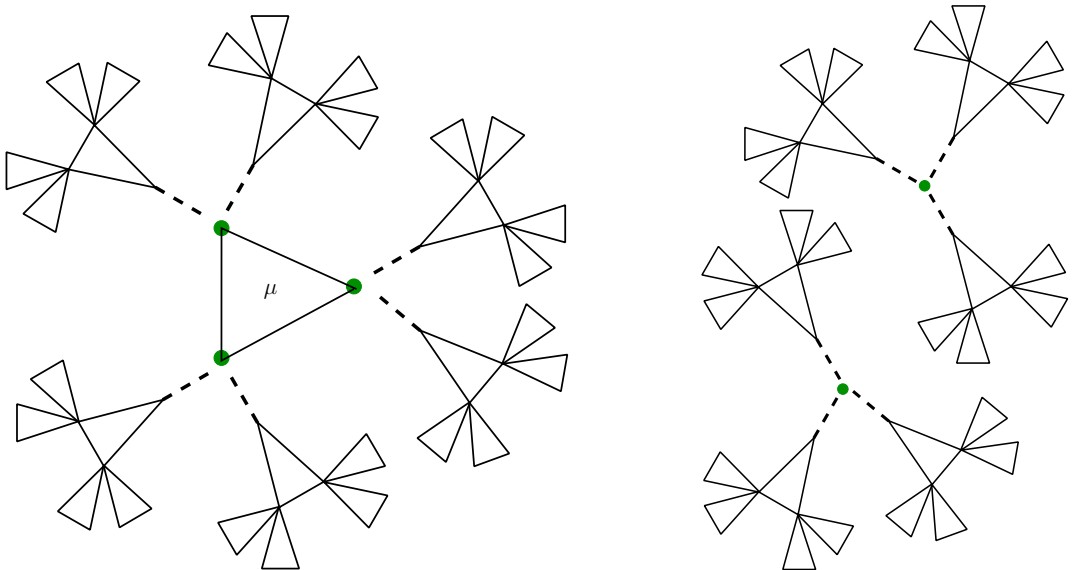

Figure 5: One can merge $(p-1)(c-1)$ sub-trees of $N$ sites either to one plaquette $\mu$ or to $(p-p/c)$ sites. In the former case, one has a new tree with $p(c-1)N+p$ sites and in the latter case, $(p-p/c)$ trees with $cN+1$ sites. We illustrate here the TPM where $c = p = 3$.

Correspondingly, the average free energy per site can be computed as the difference between a "plaquette"$(p)$ contribution and a "site" $(s)$ contribution:

$$
\begin{aligned}
-\beta f = &\frac{c}{p} \sum_{S_\mu=\pm 1} p[S_\mu] \int \prod_{j=1}^{c-1} \prod_{\nu=1}^{p} \left[ dh_\nu^{(j)} P^*(h_\nu^{(j)}) \right] \log Z_p^{(\mu)}\big(\{h_\nu^{(j)}\}, S_\mu\big) \\
&- (c-1) \int \prod_{j=1}^{c} \left[ dh_\nu^{(j)} P^*(h_\nu^{(j)}) \right] \log Z_s^{(j)}\big(\{h_\nu^{(j)}\}\big)
\end{aligned}
\tag{29}
$$

with $P^*(h)$ the sationnary probability distribution solution of Eq. (27).

The plaquette and site contributions $Z_p^{(\mu)}$ and $Z_s^{(j)}$ read respectively:

$$Z_p^{(\mu)}\big(\{h_\nu^{(j)}\}, S_\mu\big) = \sum_{\{q_\mu^{(1)},\dots,q_\mu^{(p)}=\pm 1\}} \exp\left[ \frac{\beta J S_\mu}{2} \prod_{i\in\mu} q_\mu^{(i)} + \beta \sum_{i\in\mu} q_\mu^{(i)}(\epsilon + H_\mu^{(i)}) \right] \tag{30}$$

and

$$Z_s\big(\{h_\nu^{(j)}\}\big) = \sum_{q_j=\pm 1} \exp\left[ \beta q_j(\epsilon + \sum_{\nu \ni j} h_\nu^{(j)}) \right]. \tag{31}$$

In the annealed setting, the free energy is simply expressed as

$$-\beta f = \frac{c}{p} \log Z_p(h^*) - (c-1) \log Z_s(h^*) \tag{32}$$

with $h^*$ the solution of Eq. (28) and

$$
\begin{aligned}
Z_p(h) &= \sum_{\{q_\mu^{(1)},\dots,q_\mu^{(p)}=\pm 1\}} \exp\Big[\frac{\beta \widetilde{J}}{2} \prod_{i \in \mu} q_\mu^{(i)} + \beta(\epsilon + h) \sum_{i \in \mu} q_\mu^{(i)}\Big] \\
Z_s(h) &= \sum_{q=\pm 1} \exp\big[\beta q(\epsilon + h)\big].
\end{aligned}
\tag{33}
$$

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
