# Peer review of "Role of fluctuations in the phase transitions of coupled plaquette spin models of glasses"

_SciPost Physics, doi:SciPost Phys. 1, 007 (2016)_

## Round 1 · Referee Report · Theo Nieuwenhuizen (Referee 1) · 2016-9-25

Strengths

1-analytical phase diagrams
2-numerics on Bethe and Euclidean lattices
3-sound analysis

Weaknesses

none

Report

In spin glass theory there exists the debate about the mean field vs droplet character of the phase. In general first order transitions mean field is performing well. In the present paper the authors study models for the glass transition. They have no disorder, but a slow dynamics, which should lead to a Random First Order Transition as happens in p-spin glasses.

In the study the authors consider plaquette spin models in 3d in the most interesting case c=p and compare the result to numerics on 3d Euclidean lattices. The overlap between replicas of the system is introduced, at the annealed and quenched level, which plays the role of an order parameter and allow to describe phase diagrams. Long and short range fluctuations are defined and it is argued that the latter do not change the phase diagram much when comparing Bethe lattices to Euclidean lattices. Hence MFT works qualitatively well at the static level, at least in d=3. Consequences for the dynamics are proposed.

I consider the analysis as sound and convincing and advise to publish the submitted manuscript.

Requested changes

none

  • validity: good
  • significance: high
  • originality: high
  • clarity: good
  • formatting: excellent
  • grammar: excellent

Author:  Charlotte Rulquin  on 2016-10-11  [id 63]

(in reply to Report 1 by Theo Nieuwenhuizen on 2016-09-25)

We thank the referee for the careful reading of the manuscript and the positive report.

---

## Round 1 · Referee Report · Anonymous (Referee 2) · 2016-9-29

Strengths

  1. The paper is well written
  2. An important problem is addressed: what remains of the static glass transition predicted in mean-field theory moving to finite dimensional systems
  3. Existing literature is considered in an exaustive way
  4. The main results are interesting: the mean-field nature of the transition is not washed out by long-range fluctuations in 3D.

Weaknesses

  1. No real weak points.

Report

By means of the cavity method, the authors investigate two plaquette models, plane triangular (TPM) and square pyramid (SPyM), on several graphs. Plaquettes with an odd number of sites are considered, p=3,5.
The computations are carried out coupling two replicas, both in the annealed approximation and in the quenched case. The phase diagrams are reported in the variables temperature (T) vs replica coupling constant (epsilon) and they are compared to the phase diagrams of the corresponding 3D systems.
The authors quite neatly show that, for c=p, a singular behavior of the transition line arises in the TPM and SPyM, with a transition temperature that goes to zero when the coupling goes to zero. The same behavior is found in finite dimensions and in the Bethe-lattice versions of the TPM and SPyM. In this way, the singularity is proved not to be the consequence of long-range fluctuations, that are present in 3D systems but not in mean-field. Therefore, this singularity is not an intrinsic property of finite dimensions and plaquette spin models with c = p appear to share similar glassy features on Bethe and Euclidean lattices, even at the dynamical level.

The authors find that the property of having a zero Kauzmann temperature at epsilon = 0 is a specific feature of the case c=p and it is not valid otherwise. For c > p it is T_K > 0 and T_K is absent for c < p (no phase transition, not even at zero T). A precise account of the local environment (schematized by the number of local connections c) is therefore required to recover the main features of the SPyM phase diagram. Short-range fluctuations are determinant, disregarding dimensionality.

I believe that the paper is scientifically sound, the techniques employed clearly exposed and reproducible, the (vast) literature satisfactorily acknowledged and the results interesting. I therefore recommend the paper for publication on SciPost almost in the present form.
I only ask the authors to consider a short list of changes to be made before final publication.

Requested changes

  1. In Eq. (5) \beta \epsilon -> -\epsilon

  2. Hamiltonian Eq. (7) formally corresponds to a noninteracting Ising model in an external field J/2. However, variables S are correlated because the value of the spins S in c connected plaquettes depends on the value of the same site spin \sigma they share. This correlation disappears in the dual representation when c=p. E. g., in Eq. (11) the trace does not take into account that S’s are correlated at all. The authors should briefly recall how this comes about when passing from the site to the plaquette spin variables.

  3. Soon after Eq. (14) the argument concerning the fluctuations is correct but what the authors write is not the variance, nor the covariance, but the disconnected correlation function.

  4. When recalling the duality properties the authors should spend two lines summarizing when and why duality occurs (annealed, p=c) and when it does not (quenched, annealed p\neq c).

  5. The property that hyper-plaquette models with p=c odd are fragile glasses and models with p=c even are strong glasses is an observation or there is a theoretical criterion to explain it?

  • validity: top
  • significance: top
  • originality: top
  • clarity: top
  • formatting: excellent
  • grammar: good

Author:  Charlotte Rulquin  on 2016-10-12  [id 64]

(in reply to Report 2 on 2016-09-29)

We thank the referee for the attentive reading of the manuscript and her/his nice report. Concerning the points raised by the referee, the following changes have been made:

  1. The modification is done.

  2. We thank the referee for her/his comment. Revisions have been implemented in the main text (p.7, section 3.1) on the boundary conditions allowing the one-to-one mapping, and on the consequence for $c$ plaquettes to share the same spin.

"As shown for instance in Refs. \cite{newman99,garrahan00,ritort-sollich,garrahan_annealed}, the mapping from one representation to the other is one-to-one with periodic boundary conditions (at least for the TPM and SPyM studied here). The correspondence is not exactly one-to-one for others boundary conditions, but is recovered in the thermodynamic limit.

In terms of the plaquette variables, one can reexpress the Hamiltonian in Eq. (\ref{eq_hamiltonian}) as [...] which corresponds to a noninteracting Ising model in an external field $J/2$. As is well-known\cite{newman99, garrahan00, garrahan02,ritort-sollich}, the dynamics is nonetheless glassy and the single-spin flip dynamics maps onto a relaxation with kinetic constraints for the plaquette variables. The fact that $c$ plaquettes are connected to one and the same spin leads to this nontrivial dynamics."

  1. In the revised version the squared mean is subtracted in order to recover the variance.

  2. We thank the referee for her/his comment. We add a sentence at the end of section 3.1.2. "To summarize, self-duality holds only in the annealed case for c=p.”

  3. We stated that in three dimensions hyper-plaquette models with $p = c = 5$ behave as fragile glasses and models with $p = c = 8$ behave as strong glasses. This is based on numerical simulations and previous analytical results~\cite{Rittort, Sollich, Lipowski}. However, we do not know if the general theoretical criterion suggested by the referee holds. Thus we prefer to avoid addressing this issue in the main text.

---

## Round 3 · Author Response

In this new version we implemented the modifications asked by the referee. On the 5 points raised by her/him, 4 resulted in revisions in the main text of the manuscript. For the last one, we gave an answer to the referee on the submission page.

---

## Round 3 · List of Changes

Concerning the points raised by the referee, the following changes have been made:

  1. Section 2.2, equation (5) : we changed $\beta \epsilon$ in $- \beta \epsilon$.

  2. Revisions have been implemented in the main text (p.7, section 3.1) on the boundary conditions allowing the one-to-one mapping, and on the consequence for $c$ plaquettes to share the same spin.

"As shown for instance in Refs. \cite{newman99,garrahan00,ritort-sollich,garrahan_annealed}, the mapping from one representation to the other is one-to-one with periodic boundary conditions (at least for the TPM and SPyM studied here). The correspondence is not exactly one-to-one for others boundary conditions, but is recovered in the thermodynamic limit.

In terms of the plaquette variables, one can reexpress the Hamiltonian in Eq. (???) as [...] which corresponds to a noninteracting Ising model in an external field $J/2$. As is well-known\cite{newman99, garrahan00, garrahan02,ritort-sollich}, the dynamics is nonetheless glassy and the single-spin flip dynamics maps onto a relaxation with kinetic constraints for the plaquette variables. The fact that $c$ plaquettes are connected to one and the same spin leads to this nontrivial dynamics."

  1. Soon after equation (14), the squared mean is subtracted in order to recover the variance.

  2. A sentence is added at the end of section 3.1.2. "To summarize, self-duality holds only in the annealed case for c=p.”

---

## Editorial Decision

published